# RoRecomp: Enhancing Reasoning Efficiency via Rollout Response Recomposition in Reinforcement Learning

## Abstract

Reinforcement learning with verifiable rewards (RLVR) has proven effective in eliciting complex reasoning in large language models (LLMs). However, standard RLVR training often leads to excessively verbose processes (in reasoning tasks) and inefficient exploration trajectories (in agentic settings), as outcome-only rewards provide no incentive for efficiency and the high variance in response length within relatively small rollout groups results in noisy optimization signals. To address this, we propose Rollout Response Recomposition (RoRecomp), a plug-and-play method that guides models toward concise reasoning by strategically recomposing the training data. RoRecomp separates responses into two distinct batch types: 1) priority batches, which combine the short-correct and long-incorrect responses selected from online batches to provide a clear gradient signal for brevity, and 2) compensation batches, which utilize the remaining responses stored in a replay buffer to maintain training stability and prevent model collapse. To comprehensively evaluate effectiveness, we test RoRecomp across three settings where results demonstrate substantial efficiency gains: reducing reasoning length by 27.7% in zero RL training, reducing unnecessary tool calls by 46.8% while improving accuracy in agentic RL, and achieving up to 52.5% length reduction in thinking compression, all with minimal performance impact.

## 1 Introduction

Reinforcement Learning with Verifiable Rewards (RLVR) has played a pivotal role in unlocking the complex reasoning capabilities of Large Language Models (LLMs) Team et al. (2025). By leveraging rule-based rewards, DeepSeek-R1 Guo et al. (2025) demonstrated that RL training from a base model can elicit extended chain-of-thought (CoT) reasoning and enable sophisticated cognitive behaviors. Similarly, in agentic scenarios, RLVR has enabled models to strategically employ tools multiple times to solve problems Gao et al. (2025); Jin et al. (2025). However, RLVR's reliance on outcome-based supervision is both its greatest strength and its principal limitation when optimizing for efficiency. The lack of oversight over intermediate steps may cause unnecessarily verbose thought processes in reasoning tasks or lead to excessive and redundant tool calls in agentic settings. In RLVR, the model is incentivized to explore extensively until it finds a solution, with no intrinsic penalty for verbosity, as a result, models trained with standard RLVR often exhibit progressively longer outputs. This trend is observed both when training base models to generate reasoning traces and in agentic training where the number of tool-use steps increases unnecessarily.

In principle, one might expect models to autonomously converge to an optimal response length solely from outcome reward signal, balancing the risk of "context rot" Liu et al. (2023) from overly long context in CoT against the accuracy loss from overly short ones. That is a stable operating point where the marginal utility of an extra token equals its implicit cost. However, this idealized convergence is hindered in practice by fundamental limitations of the practical RL training setup. The root cause is two-fold: a high-variance baseline estimation and an inherent algorithmic bias. First, the common practice of using a small group of samples (e.g., 8 responses per question) to estimate the reward baseline is **unbiased but with high variance**. The resulting noisy advantages inject substantial gradient variance and mask the true credit assignment for efficient CoT. Second, RL algorithms like GRPO Shao et al. (2024) have been shown to possess an inherent length bias in

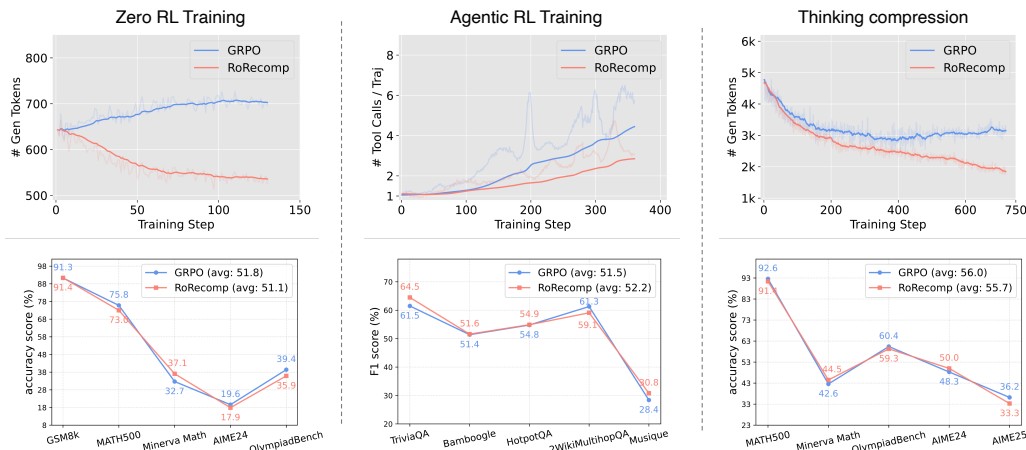

Figure 1: Comparison of RoRecomp and GRPO across three settings. (**First row**) Training dynamics demonstrate that RoRecomp significantly enhances reasoning efficiency by consistently reducing output length (Zero RL, Thinking Compression) or tool-use steps (Agentic RL). (**Second row**) This efficiency gain is achieved while maintaining **comparable performance.** Zero/Agentic RL training starts from Qwen2.5-7B; Thinking compression is trained on DeepSeek-R1-Distill-Qwen-7B.

optimization, where incorrect responses are also driven to become longer during training Liu et al. (2025). These factors combine to create conflicting and noisy optimization signals, which prevent the model from discerning truly efficient reasoning paths. Consequently, instead of converging to an optimum, the training process systematically drifts towards verbosity.

Although algorithmic length bias can be corrected with straightforward fixes, the intrinsic high variance of advantage estimation remains a fundamental challenge, as it obscures the direct credit assignment necessary to reinforce efficient reasoning behaviors. To break this cycle, we propose Rollout Response Recomposition (RoRecomp), a method that guides the model towards efficiency by strategically recomposing the data used for policy updates. RoRecomp operates after the rollout phase by reorganizing sampled responses into specialized batches. Crucially, instead of using randomly mixed samples, it constructs priority batches comprised exclusively of the most informative responses from across all questions, specifically, those that are both short and correct, or long and incorrect. This composition does not alter the advantage calculation for individual responses but fundamentally shifts the distribution of experiences presented to the optimizer in a single update step. By concentrating gradient updates on these contrasting examples, RoRecomp steers the policy more directly toward concise correctness and away from verbose errors. To maintain stability and prevent collapse, a replay buffer stores the remaining responses for occasional training in compensation batches. A dynamic learning schedule that gradually reduces the frequency of these compensation updates further refines the model's ability to balance brevity and accuracy.

Currently, reward shaping methods Hou et al. (2025); Aggarwal & Welleck (2025); Team et al. (2025) have been proposed to improve reasoning efficiency. In contrast to explicit reward shaping approaches, RoRecomp takes an orthogonal yet complementary direction. Theoretically, reward shaping methods must strictly satisfy potential-based reward shaping rules to guarantee policy invariance with respect to the original outcome objective Ng et al. (1999). In practice, however, modifying the reward function often demands delicate calibration and may still introduce unintended effects, such as oversensitivity to sequence length or deterioration in reasoning quality. RoRecomp sidesteps these issues by intervening at the level of data composition rather than altering the reward itself. By strategically recomposing the batches used for policy updates, RoRecomp implicitly guides the model towards efficiency without altering the fundamental reward. We demonstrate our versatility by combining it with a truncation penalty, where responses exceeding a length limit receive zero reward, and show that it further reduces response length beyond what reward shaping alone achieves.

The proposed method is evaluated across three practical scenarios to demonstrate its broad applicability. In the *zero RL training* setting, where RL is applied from base models to incentivize efficient reasoning, we examine whether RoRecomp achieves an optimal balance between reasoning depth and length, following the R1-zero paradigm Guo et al. (2025). In *agentic RL training*, which equips LLMs

with strategic tool-use capabilities for long-horizon tasks, we assess whether RoRecomp enhances search efficiency by reducing redundant or unproductive tool calls in information-seeking scenarios. Finally, in *RL for thinking compression*, we investigate RoRecomp's ability to effectively compress the verbose reasoning processes of off-the-shelf reasoning models, further improving their token efficiency. Experiments across three scenarios demonstrate RoRecomp's effectiveness compared to the GRPO baseline: in zero RL training, it reduces reasoning length by 27.7% with minimal accuracy drop (45.5% vs 45.9%); in agentic RL, it improves F1 score (52.2% vs 51.5%) while cutting tool calls by 46.8%; and in thinking compression, it achieves up to 52.5% length reduction while maintaining competitive performance across model scales.

## 2 RELATED WORK

**Reinforcement Learning for LLMs.** Reinforcement learning (RL) has emerged as a powerful fine-tuning method for enhancing the reasoning capacity of LLMs Jaech et al. (2024). DeepSeek-R1 Guo et al. (2025) demonstrates that pure RL can directly incentivize strong reasoning capacities in pre-trained models, underscoring the growing significance of RL in complex reasoning tasks. Among RL algorithms, Proximal Policy Optimization (PPO) Schulman et al. (2017) is widely used for reinforcement learning from human feedback, and several variants such as RLOO Ahmadian et al. (2024), GRPO Shao et al. (2024) and Reinforce++ Hu (2025)) simplify PPO and reduce computation overhead. Recently, the application of these RL algorithms to reasoning tasks has advanced rapidly. For instance, DAPO Yu et al. (2025) accelerates model convergence by filtering zero-gradient examples; VC-PPO Yuan et al. (2025b) investigates the causes of PPO collapse in long CoT settings and proposes techniques to stabilize long CoT training; and VAPO Yuan et al. (2025a) introduces length-adaptive GAE to optimize advantage estimation for long CoT responses; While these methods primarily aim to enhance reasoning by encouraging longer responses, our work instead leverages RL to compress the CoT of strong long-CoT models, seeking to maintain reasoning performance while reducing response length.

**Reasoning Compression in LLMs.** Efficient reasoning compression aims to achieve System 1 speed while retaining System 2 performance Snell et al. (2024). Existing methods fall into training-free and optimization-based categories. Training-free approaches include prompt engineering Xu et al. (2025), decoding-time interventions Muennighoff et al. (2025), and model merging Wu et al. (2025); Team et al. (2025). While effective in reducing length, these methods are orthogonal to our RL-based approach and can be combined for further gains.

Optimization-based methods are further divided into offline and online approaches. Offline methods Xia et al. (2025); Luo et al. (2025a); Shen et al. (2025) use concise CoT trajectories for SFT or preference learning (e.g., DPO). Online RL methods directly optimize length during training: Kimi-1.5 Team et al. (2025) adds length penalty rewards; ConciseRL Fatemi et al. (2025) selects solvable data for PPO; ThinkPrune Hou et al. (2025) iteratively tightens length constraints in GRPO. Our method belongs to this category and is compared with these approaches in Sec. 4.2.

## 3 METHOD: ROLLOUT RESPONSE RECOMPOSITION

In this section, we first introduce the background knowledge of standard RL frameworks. Then we introduce how we recompose rollout responses into the priority batch and compensation batch.

### 3.1 PRELIMINARY

Reinforcement Learning (RL) for LLMs follows an iterative two-stage process comprising response generation and policy optimization Ouyang et al. (2022). During the sampling phase, the actor generates multiple diverse responses for each input prompt. The subsequent training phase leverages the reward signals of each response to update policy model through gradient-based optimization, employing mechanism like PPO Schulman et al. (2017) and GRPO Shao et al. (2024).

**Verifiable Rewards.** RL with verifiable rewards (RLVR) plays a vital role in incentivizing reasoning capability Guo et al. (2025); Team et al. (2025). It offers precise reward signals, reducing the risk of reward hacking. For math and coding questions, outputs from the policy model are evaluated by a verifier $\mathcal{V}$. Specifically, in the present study, we investigate both the maths and agent tasks. For

Figure 2: (a) The overall framework of RoRecomp. After the response generation, we recompose candidate responses into two types of batches: priority batches and compensation batches. (b) The details of response selection. We select prioritized responses for each question by jointly considering the response length and reward.

mathematics, we assign a reward of 1 only if both the answer and its wrapped format are correct via exact match; otherwise, the reward is 0. In agentic RL, we consider the information seeking scenario where LLMs are equipped with tools (e.g., search) to access external knowledge base for question answering. The F1 score between the prediction and the reference answer is used as the reward signal. A binary format reward is employed to ensure adherence to the ReAct Yao et al. (2023) paradigm.

**Proximal Policy Optimization (PPO).** PPO Schulman et al. (2017) is a classical actor-critic RL algorithm, which uses a critic model to serve as value function to estimate the value for each token in outputs. To ensure stable learning, token-wise KL divergence between the current policy and reference model is calculated and integrated into the rewards. Combing the predicted rewards and values, PPO uses the Generalized Advantage Estimation (GAE) to calculate advantages $\hat{A}_t$ for each token. The policy model $\pi_\theta$ is optimized by maximum the following objective:

$$\mathcal{J}_{\text{PPO}} = \mathbb{E}_{q \sim D, o \sim \pi_{\theta_{old}}} \frac{1}{|o|} \sum_{t=1}^{|o|} \left[ \min\left( \frac{\pi_\theta(o_t|q, o_{\leq t})}{\pi_{\theta_{old}}(o_t|q, o_{\leq t})} \hat{A}_t, \text{clip}\left( \frac{\pi_\theta(o_t|q, o_{\leq t})}{\pi_{\theta_{old}}(o_t|q, o_{\leq t})}, 1 - \epsilon, 1 + \epsilon \right) \hat{A}_t \right) \right]$$

(1)

where $\epsilon$ is clipping ranging of the importance sampling ratio, $q$ refers to the input question, and $o$ is the output sampled from the old policy model $\pi_{\theta_{old}}$.

**Group Relative Policy Optimization (GRPO).** To reduce computational overhead, GRPO Shao et al. (2024) eliminates the critic model, which is typically comparable in size to the policy model and requires separate updates during training. Instead, it approximates the value function using the group mean reward as a baseline. Specifically, for a group of outputs ($\{o_i\}_{i=1}^G$) sampled from the same question, their rewards $\mathbf{r} = \{r_i\}_{i=1}^G$ are normalized within the group to obtain advantages: $\hat{A}_i = \frac{r_i - mean(\mathbf{r})}{std(\mathbf{r})}$. In this paper, we use PPO and GRPO as the default RL frameworks. For GRPO implementation, we adopt the normalization term modification from Liu et al. (2025) to mitigate inherent length bias in the objective function.

### 3.2 FORMATION OF PRIORITY AND COMPENSATION BATCHES

Empirical studies of large-scale RLVR training, both in *zero RL* and *agentic RL* settings, have consistently observed a trend of increasing response length Zeng et al. (2025); Gao et al. (2025). This extended thinking process often encompasses beneficial reasoning behaviors like self-reflection and self-critique. However, guided solely by outcome reward models (ORM) without intermediate efficiency supervision, the resulting reasoning processes can be highly suboptimal. The severity of this issue is exemplified by the high variance in response length observed in practice. For instance, when sampling from DeepSeek-R1 Guo et al. (2025) on AIME24 MAA (2024), we observe an average discrepancy of 8.3k tokens between the longest and shortest responses for the same problem. In standard RLVR frameworks, the advantage baseline is computed within relatively small rollout groups (typically 8-16 responses per prompt) due to computational constraints. Such combination of high length variance with a small group size results in noisy advantage estimates that fail to provide a clear signal for distinguishing efficient from verbose reasoning paths. RoRecomp addresses this core issue by recomposing the training data to provide a policy gradient signal that explicitly rewards reasoning efficiency.

**Priority Batch as a Modulator.**     We depict the framework of RoRecomp in Fig. 2. By adjusting sampling parameters such as temperature and top-p, randomness is introduced into response genera-tion, allowing to produce multiple diverse outputs for each input prompt. This process of generating responses is referred to as the *rollout* Shao et al. (2024). After generating a set of responses $\mathcal{R}$ for each input, a rule-based reward model is employed to clarify each response as correct or incorrect, formatting two subsets: $\mathcal{R}_{\text{correct}}$ and $\mathcal{R}_{\text{incorrect}}$. Subsequently, advantages are computed using either GAE Schulman et al. (2017) or group reward normalization Shao et al. (2024). The policy model is then optimized to reinforce high-reward response patterns while suppressing low-scoring outputs. The proposed RoRecomp method operates after response generation, recomposing responses for the following policy optimization. To tile the gradient direction towards brevity, we elaborately select a subset of prioritized responses for each input question. Specifically, we select the shortest $\alpha$ fraction from $\mathcal{R}_{\text{correct}}$ and the longest $\alpha$ fraction from $\mathcal{R}_{\text{incorrect}}$:

$$\mathcal{B}_{\text{priority}} = \text{Top-}\alpha \text{ shortest in } \mathcal{R}_{\text{correct}} \cup \text{Top-}\alpha \text{ longest in } \mathcal{R}_{\text{incorrect}}, \tag{2}$$

The prioritized responses are reorganized as **priority batches** $\mathcal{B}_{\text{priority}}$, which encourages concise cor-rect reasoning while suppressing verbose errors. The remaining responses, which are of intermediate length, are stored in an experience replay buffer for deferred training. Once the buffer is full, the oldest experiences are popped to form a **compensation batch** $\mathcal{B}_{\text{comp}}$ for an additional training step.

The choice of the selection ratio (e.g., $\alpha$=80%) is a direct response to the high variance of ad-vantage estimates in small rollout groups. RoRecomp reduces this variance by filtering out the intermediate-length responses that contribute most to noisy and ambiguous learning signals. This strategy intentionally introduces a beneficial bias, focusing updates on the most contrasting examples: concise correctness and verbose errors. The value of $\alpha$ is selected to balance this variance reduction against the need for a sufficient number of priority samples to ensure stable gradient estimates. A smaller $\alpha$ value strengthens the emphasis on brevity but may lead to training instability due to limited samples, while a larger $\alpha$ provides more stable updates at the cost of reduced compression effect.

**Compensation Batch as a Regularizer.**     The alternating training between priority and compensation batches implements an implicit curriculum learning strategy. The model first focuses on mastering the core principle of efficiency by learning from the most informative samples in the priority batches. This phase emphasizes the strong correlation between response length and reward outcomes. Subsequently, the compensation batches provide a broader review of general reasoning patterns, ensuring the model maintains its fundamental capabilities while refining its efficiency. This structured learning process, ranging from focused efficiency optimization to comprehensive capability maintenance, facilitates a balance between reasoning brevity and accuracy.

To better balance efficiency and performance, we implement a dynamic schedule for compensation batches. Empirical results show that reducing the frequency of compensation batches after the model's reward stabilizes leads to shorter responses. We achieve this through a cosine decay schedule for the compensation batch probability:

$$p_{\text{comp}} = \max\left(p_{\text{lower}}, \frac{1 + \cos(\pi \cdot T_t/T_{\text{max}})}{2}\right), \tag{3}$$

where $p_{\text{lower}} = 0.2$ denotes the lower bound, $T_t$ is the current training step, and $T_{\text{max}}$ is the total number of training steps. This ensures stable learning initially while increasingly prioritizing length reduction as training progresses.

**Discussion.**     RoRecomp's effectiveness stems from recomposing the sample distribution for policy gradient estimation. While standard RLVR uses Monte Carlo sampling over random responses, RoRecomp constructs batches from distribution $P_{\text{priority}}$ that over-represents informative samples:

$$\nabla J(\theta) \approx \mathbb{E}_{r \sim P_{\text{priority}}} \left[A(r)\nabla_\theta \log \pi_\theta(r)\right]$$

The priority batch creates a biased estimator that amplifies positive advantages from short-correct responses and reinforces negative advantages from long-incorrect responses. This provides clearer optimization signals than standard batches. Compensation batches from a replay buffer serve as regularizers, maintaining reasoning capabilities while the gradual reduction of compensation updates guides stable convergence toward efficient reasoning. By recomposing data rather than modifying rewards, RoRecomp offers a more stable path to efficiency.

Table 1: Results of *zero RL training* on Qwen2.5-7B base, reporting the mean@16 accuracy ("acc") and the average response token length ("len").

| Methods | GSM8K | | MATH500 | | AIME24 | | AIME25 | | AMC23 | | Minerva | | Olympiad | | Avg. | |
|---|---|---|---|---|---|---|---|---|---|---|---|---|---|---|---|---|
| | acc | len | acc | len | acc | len | acc | len | acc | len | acc | len | acc | len | acc | len |
| *Qwen2.5-7B* | 87.7 | - | 60.5 | - | 10.0 | - | 3.3 | - | 32.8 | - | 19.5 | - | 27.8 | - | 34.5 | - |
| GRPO Baseline | 91.3 | 323 | 75.8 | 734 | 19.6 | 1361 | 3.3 | 1389 | 59.1 | 1187 | 32.7 | 957 | 39.4 | 1030 | 45.9 | 997 |
| **+ RoRecomp** | 91.2 | 245 | 73.0 | 604 | 17.9 | 1087 | 3.3 | 891 | 57.8 | 897 | 37.1 | 558 | 38.5 | 763 | 45.5 | **721** |

Table 2: Results of *agentic RL trianing* on Qwen2.5-7B base, reporting the averaged F1 score ("F1") and the number of tool calls ("# tool") per trajectory.

| Methods | TriviaQA | | Bamboogle | | HotpotQA | | 2WikiMQA | | Musique | | Avg. | |
|---|---|---|---|---|---|---|---|---|---|---|---|---|
| | F1 | # tool | F1 | # tool | F1 | # tool | F1 | # tool | F1 | # tool | F1 | # tool |
| *Qwen2.5-7B* | 50.4 | 1.5 | 37.2 | 2.0 | 29.2 | 1.5 | 30.4 | 2.2 | 11.8 | 2.1 | 31.8 | 1.9 |
| GRPO Baseline | 61.5 | 6.2 | 51.4 | 6.2 | 54.8 | 6.2 | 61.3 | 6.3 | 28.4 | 6.3 | 51.5 | 6.2 |
| **+ RoRecomp** | 64.5 | 2.8 | 51.6 | 3.4 | 54.9 | 3.2 | 59.1 | 3.4 | 30.8 | 3.5 | **52.2** | **3.3** |

## 4 EXPERIMENTS

### 4.1 EXPERIMENTAL SETTINGS

**Zero RL Training.** In this setting, we perform RL training directly on the Qwen2.5-7B base model Yang et al. (2024), following the same training protocol and dataset as SimpleRL-zoo Zeng et al. (2025). The training configuration uses a batch size of 1024 for 130 training steps. We evaluate on seven mathematical reasoning benchmarks: GSM8K Cobbe et al. (2021), AIME 2024, AIME 2025, AMC 2023, MATH-500 Lightman et al. (2023), Minerva Math Lewkowycz et al. (2022), and OlympiadBench He et al. (2024). Performance is measured by pass@1 accuracy, averaged over 16 samples per question.

**Agentic RL Training.** We train search agents following Asearcher Gao et al. (2025) with the AReal codebase Fu et al. (2025), equipping the model with a locally deployed RAG system that retrieves information from a Wikipedia 2018 corpus. The agent has access to two tools: a search engine and a web content fetcher. Training starts from the Qwen2.5-7B model on 35K training examples from Asearcher, with a maximum of 32 interaction turns allowed per episode. The training runs for 350 steps with a batch size of 64. Evaluation covers one single-hop QA benchmark (TriviaQA Joshi et al. (2017)) and four multi-hop QA benchmarks (HotpotQA Yang et al. (2018), 2WikiMultiHopQA Ho et al. (2020), MuSiQue Trivedi et al. (2022), and Bamboogle Press et al. (2022)).

**Thinking Compression on Reasoning Models.** We use DeepSeek-R1-Distill-Qwen 1.5B and 7B models Guo et al. (2025) (abbreviated as DeepSeek-1.5B/7B) as base models for compression. Both GRPO Shao et al. (2024) and PPO Schulman et al. (2017) are employed as RL frameworks, implemented using the verl codebase Sheng et al. (2024). Training uses a learning rate of 1e-6 without warmup, with a prompt batch size of 224 and 12 responses sampled per input prompt. The training data consists of 40K competition-level math questions from DeepScaleR-Preview Luo et al. (2025b), with a maximum response length of 8192 tokens. All experiments run for 720 steps. Evaluation includes mathematical reasoning benchmarks as well as LiveCodeBench (2024.08-2025.01) Jain et al. (2024) for coding and GPQA Diamond Rein et al. (2024) for scientific reasoning.

### 4.2 MAIN RESULTS

**Zero RL Training.** Table 1 presents the results of zero RL training starting from the Qwen2.5-7B base model. RoRecomp demonstrates significant improvements in reasoning efficiency while maintaining competitive accuracy across all mathematical benchmarks. Compared to the GRPO baseline, our method reduces the average response length from 997 tokens to 721 tokens (a 27.7% reduction), with only a marginal decrease in average accuracy (45.5% vs. 45.9%). Notably, on Minerva Math, RoRecomp not only reduces length by 41.7% (from 957 to 558 tokens) but also improves accuracy from 32.7% to 37.1%. These results indicate that RoRecomp effectively guides the model toward more concise reasoning without sacrificing solution quality.

The training dynamics in Fig. 3 provide further insight into this behavior. While the GRPO baseline exhibits a continuous increase in response length throughout training, which is often misinterpreted

Table 3: Results of ***thinking compression*** on reasoning models DeepSeek-1.5B/7B, reporting the mean@16 accuracy ("acc") and the average response token length ("len").

| Methods | MATH500 | | AIME24 | | AIME25 | | AMC23 | | Minerva | | Olympiad | | Avg. | |
|---|---|---|---|---|---|---|---|---|---|---|---|---|---|---|
| | acc | len | acc | len | acc | len | acc | len | acc | len | acc | len | acc | len |
| *DeepSeek-1.5B* | 83.0 | 5961 | 28.3 | 18082 | 25.8 | 17420 | 70.9 | 10295 | 31.2 | 7682 | 44.0 | 12518 | 47.0 | 11993 |
| GRPO Baseline | 86.2 | 2594 | 27.1 | 6519 | 22.5 | 6164 | 75.0 | 3919 | 34.6 | 3059 | 49.0 | 4196 | **49.1** | 4408 |
| + RoRecomp | 84.6 | 1126 | 27.9 | 3473 | 23.3 | 2860 | 74.1 | 2100 | 33.1 | 1078 | 46.4 | 1935 | 48.2 | **2095** |
| PPO Baseline | 82.4 | 2016 | 27.1 | 4805 | 19.6 | 4399 | 71.9 | 3236 | 34.6 | 2058 | 46.5 | 3270 | 47.0 | 3297 |
| + RoRecomp | 83.6 | 1435 | 28.8 | 3383 | 18.8 | 3003 | 74.4 | 1972 | 33.8 | 1256 | 46.5 | 2128 | **47.6** | **2196** |
| *DeepSeek-7B* | 92.8 | 4081 | 52.7 | 13432 | 40.4 | 14885 | 89.5 | 6575 | 42.6 | 5116 | 60.0 | 9322 | 63.0 | 8901 |
| GRPO Baseline | 92.6 | 2278 | 48.3 | 6241 | 36.2 | 6546 | 89.7 | 3423 | 42.6 | 2455 | 60.4 | 4024 | **61.6** | 4161 |
| + RoRecomp | 91.4 | 1324 | 50.0 | 3591 | 33.3 | 3539 | 86.6 | 1966 | 44.5 | 1208 | 59.3 | 2197 | 60.8 | **2304** |
| *Qwen3-8B* (non-thinking) | 83.2 | 1242 | 23.7 | 6396 | 17.9 | 5491 | 68.1 | 2446 | 32.4 | 655 | 50.4 | 2976 | 46.0 | 3201 |
| *Qwen3-8B* | 95.1 | 5402 | 73.3 | 15383 | 66.2 | 18165 | 94.4 | 9311 | 48.3 | 7072 | 68.4 | 11373 | **74.3** | 11118 |
| GRPO Baseline | 95.1 | 4037 | 72.9 | 10999 | 59.2 | 13878 | 92.5 | 6667 | 49.1 | 4910 | 68.2 | 7855 | 72.8 | 8058 |
| + RoRecomp | 94.9 | 3144 | 69.6 | 8274 | 56.2 | 9571 | 94.7 | 4983 | 60.2 | 3701 | 65.8 | 5843 | **73.6** | **5929** |

as the emergence of beneficial cognitive behaviors like self-reflection, RoRecomp demonstrates that such length growth is not necessarily correlated with improved performance. Our method achieves comparable final rewards while stabilizing output length at a significantly lower level. This indicates that the lengthy exploration in standard RLVR is often inefficient. RoRecomp successfully steers the exploration process itself toward more concise reasoning.

**Agentic RL Training.** The agentic RL training results in Table 2 show that RoRecomp achieves a better balance between task performance and operational efficiency. Our method increases the average F1 score from 51.5% to 52.2% while reducing the average number of tool calls per trajectory from 6.2 to 3.3 (a 46.8% reduction). This efficiency improvement is consistent across both single-hop and multi-hop QA benchmarks, demonstrating RoRecomp's ability to adaptively adjust tool-usage strategies based on task complexity. On the simpler single-hop task (TriviaQA), RoRecomp improves the F1 score from 61.5% to 64.5% while significantly reducing the average number of tool calls from 6.2 to 2.8. For more complex multi-hop tasks (Bamboogle, HotpotQA), it maintains comparable F1 scores while cutting tool calls by nearly half. These results indicate that RoRecomp guides the model toward more focused and efficient tool usage, eliminating unnecessary steps without compromising answer quality.

**Thinking Compression on Reasoning Models.** The comprehensive results on thinking compression are presented in Table 3. All methods are trained with a maximum generation length of 8k tokens, which acts as an implicit reward shaping mechanism by truncating longer responses. This explains why the GRPO baseline itself achieves significant compression compared to the original models. Beyond this baseline effect, RoRecomp demonstrates remarkable effectiveness in further compressing the verbose reasoning processes of off-the-shelf models across different scales and RL backbones, consistently achieving drastic length reductions while preserving competitive performance.

For the DeepSeek-1.5B model, RoRecomp reduces the average response length by 52.5% (from 4,408 to 2,095 tokens) when applied with GRPO, with a minimal accuracy drop of 0.9 points (49.1% to 48.2%). A similar trend is observed with PPO, where length is reduced by 33.4% with a slight performance improvement. On the larger DeepSeek-7B model, based on GRPO, our RoRecomp cuts the average length nearly in half (from 4,161 to 2,304 tokens, a 44.6% reduction) with an accuracy drop of only 0.8 points. Most notably, on the strong Qwen3-8B model Yang et al. (2025) in thinking mode, RoRecomp achieves a 26.4% length reduction (from 8,058 to 5,929 tokens) while marginally improving the average accuracy from 72.8% to 73.6%. These results underscore the generality of RoRecomp as a plug-and-play method for enhancing reasoning efficiency without compromising the problem-solving capabilities of powerful reasoning models.

**Generalization on out-of-domain benchmarks.** Although our models are trained solely on mathematical data, we further evaluate their generalization ability on coding (LiveCodeBench) and science reasoning (GPQA) tasks, which represent out-of-domain scenarios. As shown in Tab. 4, RoRecomp consistently reduces response length on these OOD test sets. For DeepSeek-1.5B models, RoRecomp not only surpasses the vanilla GRPO/PPO baseline and original DeepSeek models in accuracy, but also generates much shorter responses; for example, with GRPO, RoRecomp reduces the average response length by 32% (from 7944 to 5416 tokens). For DeepSeek-7B models, RoRecomp continues to effectively compress the output length, though with a slight drop in accuracy.

Table 4: Evaluation on out-of-domain testsets.

| Methods | GPQA | | LiveCodeBench | | Avg. | |
|---|---|---|---|---|---|---|
| | acc | len | acc | len | acc | len |
| *DeepSeek-1.5B* | 36.4 | 18324 | 17.9 | 15057 | 27.2 | 16690 |
| GRPO Baseline | 38.4 | 6001 | 16.8 | 9886 | 27.6 | 7944 |
| + RoRecomp | 39.9 | 4067 | 20.5 | 6766 | **30.2** | **5416** |
| PPO Baseline | 34.8 | 5501 | 17.5 | 8242 | 26.2 | 6872 |
| + RoRecomp | 36.4 | 4396 | 16.8 | 6888 | **26.6** | **5642** |
| *DeepSeek-7B* | 53.5 | 7985 | 37.3 | 12978 | 45.4 | 10482 |
| GRPO Baseline | 51.5 | 4718 | 37.9 | 7721 | **44.7** | 6220 |
| + RoRecomp | 48.5 | 3817 | 38.4 | 6070 | 43.4 | **4944** |

Table 5: Comparison with concurrent reasoning compression methods: ThinkPrune Hou et al. (2025), ConciseRL Fatemi et al. (2025), and AdaR1 Luo et al. (2025a).

| Method | MATH500 | | AIME24 | | AIME25 | | AMC23 | | Olympiad | | Avg. | |
|---|---|---|---|---|---|---|---|---|---|---|---|---|
| | acc | len | acc | len | acc | len | acc | len | acc | len | acc | len |
| *DeepSeek-1.5B* | | | | | | | | | | | | |
| ThinkPrune | 83.2 | 1938 | 27.1 | 5631 | - | - | 73.2 | 3039 | - | - | 61.2 | 3536 |
| ConciseRL | 81.0 | 1965 | 30.0 | 6752 | - | - | 69.4 | 2936 | - | - | 60.1 | 3884 |
| **RoRecomp** | 84.6 | 1126 | 27.9 | 3473 | - | - | 74.1 | 2100 | - | - | **62.2** | **2233** |
| AdaR1 | 80.8 | 2455 | - | - | 23.0 | 9516 | - | - | 42.1 | 5802 | 48.6 | 5924 |
| **RoRecomp** | 84.6 | 1126 | - | - | 18.8 | 3003 | - | - | 46.4 | 1935 | **49.9** | **2021** |
| *DeepSeek-7B* | | | | | | | | | | | | |
| AdaR1 | 90.2 | 1468 | - | - | 35.8 | 8426 | - | - | 52.4 | 4889 | 59.5 | 4928 |
| **RoRecomp** | 91.4 | 1324 | - | - | 33.3 | 3539 | - | - | 59.3 | 2197 | **61.3** | **2353** |

Table 6: Response length and pass@1 scores across 5 MATH subsets with varying difficulty levels.

| Methods | Difficulty Level | | | | |
|---|---|---|---|---|---|
| | Level 1 | Level 2 | Level 3 | Level 4 | Level 5 |
| **Response length (tokens)** | | | | | |
| *DeepSeek-1.5B* | 2587 | 3130 | 3903 | 4903 | 7082 |
| + RoRecomp | 495 (-81%) | 647 (-79%) | 825 (-79%) | 1121 (-77%) | 1606 (-77%) |
| **Accuracy (mean@16)** | | | | | |
| *DeepSeek-1.5B* | 93.7 | 92.0 | 88.6 | 84.6 | 71.9 |
| + RoRecomp | 94.4 | 93.1 | 90.0 | 85.4 | 74.3 |

Table 7: Ablation study on the effect of $\alpha$ across math and out-of-domain benchmarks. Each entry shows "accuracy [response length]".

| $\alpha$ | Math (Avg) | GPQA | LiveCodeBench |
|---|---|---|---|
| 0.5 | 40.1 [921] | 35.1 [2582] | 15.5 [4910] |
| 0.7 | 48.0 [1711] | 39.9 [3605] | 18.3 [5922] |
| 0.8 | 48.2 [2095] | 39.9 [4067] | 20.5 [6766] |
| 0.9 | 49.3 [2979] | 38.5 [4742] | 19.2 [7262] |

**Comparison with concurrent works.** Recently, severe works have been proposed to enhance reasoning efficiency, some of which also adopt DeepSeek models as their base, enabling fair comparisons with our approach. Methods presented in Tab. 5 employ online reinforcement learning techniques. Specifically, ThinkPrune Hou et al. (2025) iteratively reduces the generation limit from 4k to 2k during the GRPO training; ConciseRL Fatemi et al. (2025) selects a limited set of problems that are at least occasionally solvable as train data and uses PPO for optimization; While our RoRecomp also utilizes GRPO. The results indicate that RoRecomp consistently surpasses these two methods in both accuracy and response length. For example, RoRecomp achieves an average accuracy of 62.2%, surpassing ThinkPrune's 61.2%, while reducing the average response length from 3536 to 2233 tokens. AdaR1 leverages collected preference pairs and DPO Rafailov et al. (2023) to improve reasoning efficiency. RoRecomp consistently achieves higher accuracy and significantly shorter responses than AdaR1. Furthermore, as an online RL method, RoRecomp is simpler to implement, as it does not require meticulous offline data collection.

## 4.3 ABLATION STUDY

Ablation studies are conducted under the *thinking compression* setting using DeepSeek-R1-Distill-Qwen-1.5B with GRPO. This setup provides a clear testbed for evaluating reasoning length reduction, as it involves compressing the verbose reasoning traces of a off-the-shelf reasoning model.

**RoRecomp's Compression Effect across Difficulty Levels.** To address whether RoRecomp truly compresses reasoning length rather than simply distinguishing between easy and hard questions, we report both response length and accuracy across the five difficulty levels in the MATH benchmark Lightman et al. (2023). As difficulty increases from level 1 to 5, RoRecomp consistently reduces response length by around 80% at each level, while slightly improving pass@1 accuracy over the original DeepSeek-R1-Distill-Qwen model. Results are depicted in Tab. 6.

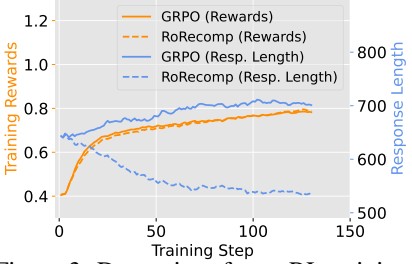

Figure 3: Dynamics of zero RL training.

Table 8: Results of pass@1 and pass@32 accuracy on math benchmarks.

| Methods | MATH500 | AIME24 | AIME25 | AMC23 | Minerva | Olympiad | Avg. |
|---|---|---|---|---|---|---|---|
| *DeepSeek-1.5B* | | | | | | | |
| Pass@1 | 82.2 | 26.7 | 19.6 | 68.1 | 30.1 | 44.4 | 45.2 |
| + RoRecomp | 84.6 | 27.9 | 23.3 | 74.1 | 33.1 | 46.4 | 48.2 (+3.0) |
| Pass@32 | 96.4 | 73.3 | 53.3 | 92.5 | 55.1 | 72.4 | 74.1 |
| + RoRecomp | 95.8 | 70.0 | 53.3 | 95.0 | 52.9 | 69.3 | 72.7 (-1.4) |
| *DeepSeek-7B* | | | | | | | |
| Pass@1 | 92.0 | 51.7 | 38.3 | 88.7 | 41.9 | 58.2 | 61.8 |
| + RoRecomp | 91.4 | 50.0 | 33.3 | 86.6 | 44.4 | 59.3 | 60.8 (-1.0) |
| Pass@32 | 98.0 | 80.0 | 70.0 | 97.5 | 62.9 | 76.1 | 80.8 |
| + RoRecomp | 97.8 | 80.0 | 66.7 | 97.5 | 58.5 | 73.5 | 79.0 (-1.8) |

Table 9: Comparison of RoRecomp with length penalty methods under different training token budget.

| Method | Budget | MATH500 | | AIME24 | | AIME25 | | AMC23 | | Minerva | | Olympiad | | Average | |
|--------|--------|---------|-----|--------|-----|--------|-----|-------|-----|---------|-----|----------|-----|---------|-----|
| | | acc | len | acc | len | acc | len | acc | len | acc | len | acc | len | acc | len |
| GRPO Baseline | 16K | 85.0 | 4260 | 28.8 | 9894 | 23.3 | 9710 | 77.2 | 6698 | 32.0 | 5091 | 46.9 | 7451 | 48.9 | 7184 |
| Length Penalty (kimi) | 16K | 86.9 | 3039 | 30.4 | 8436 | 20.8 | 7909 | 76.2 | 5289 | 31.4 | 3454 | 47.8 | 5733 | 48.9 | 5643 |
| RoRecomp (Ours) | 16K | 86.7 | 1894 | 28.3 | 5728 | 21.6 | 4800 | 73.8 | 3018 | 31.2 | 1705 | 48.4 | 3351 | 48.3 | **3416** |
| GRPO Baseline | 8K | 92.6 | 2278 | 48.3 | 6241 | 36.2 | 6546 | 89.7 | 3423 | 42.6 | 2455 | 60.4 | 4024 | **61.6** | 4161 |
| Length Penalty (kimi) | 8K | 86.0 | 2872 | 28.8 | 6892 | 22.5 | 6219 | 75.0 | 4641 | 32.4 | 3349 | 49.5 | 4728 | 49.0 | 4783 |
| RoRecomp (Ours) | 8K | 84.6 | 1126 | 27.9 | 3473 | 23.3 | 2860 | 74.1 | 2100 | 33.1 | 1078 | 46.4 | 1935 | 48.2 | **2095** |

**Sensitivity of Selection Ratio $\alpha$.** We analyze the impact of the selection ratio $\alpha$, which determines the fraction of responses included in the priority batch. As shown in Table 7, smaller $\alpha$ values (e.g., 0.5) prioritize the most contrasting examples, yielding the shortest responses but lowest accuracy. Larger values (e.g., 0.9) include more medium-length responses, improving accuracy at the cost of increased length. The optimal balance is achieved at $\alpha = 0.8$, maintaining competitive accuracy (48.2% on math) while substantially reducing response length (2095 tokens). This setting also generalizes well to out-of-domain tasks. The minimal performance variation between $\alpha = 0.7$ and 0.8 demonstrates robustness to parameter tuning.

**Effect on Sampling Diversity.** We analyze the impact of RoRecomp on sampling diversity by comparing pass@1 and pass@32 performance in Table 8. RoRecomp improves or maintains pass@1 accuracy across model scales while achieving a minimal reduction in pass@32 scores (only -1.4 and -1.8 points for 1.5B and 7B models, respectively). This indicates that our method effectively compresses reasoning length while marginally affecting the diversity of valid solutions. The preserved pass@1 performance demonstrates maintained problem-solving capability, and the negligible pass@32 change confirms that compression primarily eliminates redundant paths without substantially limiting the model's ability to generate diverse reasoning trajectories.

**Comparison with Length Penalty Reward Shaping.** We conduct a comprehensive comparison between RoRecomp and the competitive length penalty reward shaping approach Team et al. (2025) under different training-time maximum generation length settings. As shown in Table 9, when trained with a 16K token limit (where the truncation penalty is weaker), the explicit length penalty reduces average response length by 1,541 tokens compared to the GRPO baseline (from 7,184 to 5,643 tokens), while RoRecomp achieves a more substantial reduction of 3,768 tokens. This demonstrates that RoRecomp provides superior length compression even under relaxed constraints.

More importantly, when both methods are trained with an 8K token limit, which itself acts as an implicit reward shaping mechanism by truncating responses exceeding this length and assigning zero reward, RoRecomp achieves significantly shorter outputs (2,095 tokens) compared to the length penalty approach (4,783 tokens). This performance gap arises because the explicit length penalty functionally overlaps with this implicit reward shaping, diminishing its additional effect. In contrast, RoRecomp operates orthogonally through data recomposition rather than reward modification, allowing it to synergize effectively with the truncation-based reward shaping. The results confirm that RoRecomp provides a fundamentally different and more effective approach to reasoning compression compared to explicit reward shaping methods.

## 5 CONCLUSION

This work addresses the critical problem of verbose reasoning in RL with Verifiable Rewards (RLVR) through Rollout Response Recomposition (RoRecomp), a plug-and-play method that guides models toward efficiency via strategic data recomposition rather than reward modification. By separating responses into priority batches (emphasizing concise correctness) and compensation batches (ensuring stability), RoRecomp provides clearer optimization signals for efficient reasoning. Comprehensive experiments across zero RL training, agentic RL, and thinking compression demonstrate RoRecomp's effectiveness: it reduces reasoning length by up to 74% and tool calls by 46.8% with minimal performance impact, outperforming reward-shaping baselines. Our approach highlights data composition as a powerful lever for efficiency optimization, offering a simpler and more stable alternative to reward engineering for building concise yet capable reasoning models.

**Ethics Statement**  The present study conforms with the ICLR Code of Ethics. The paper does not involve crowdsourcing nor research with human subjects.

**Reproducibility Statement**  All datasets used in the paper are publicly accessible (see Section 4.1). All the codes are available at supplementary materials for reproduction. In addition, we provide all the details of implementation in Section 4.1.

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

Table 10: Comparing the number of steps and tokens in different reasoning phases before and after applying RoRecomp.

| Methods | Step Count | | | Token Count | | |
|---|---|---|---|---|---|---|
| | problem-understanding | problem-solving | self-verification | problem-understanding | problem-solving | self-verification |
| DeepSeek-R1-Distill-Qwen-1.5B | 12 | 410 | 55 | 670 | 11738 | 1600 |
| + RoRecomp | 7 (↓42%) | 94 (↓77%) | 10 (↓82%) | 361 (↓46%) | 2900 (↓75%) | 309 (↓81%) |
| DeepSeek-R1-Distill-Qwen-7B | 8 | 330 | 44 | 474 | 9876 | 1320 |
| + RoRecomp | 5 (↓38%) | 82 (↓75%) | 5 (↓89%) | 297 (↓37%) | 2884 (↓71%) | 217 (↓84%) |

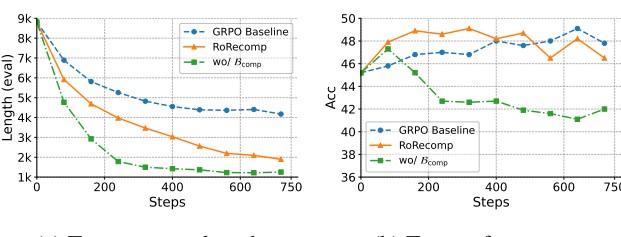

(a) Test response length.          (b) Test performance.

Figure 4: Effectiveness of compensation batches, with response length and performance averaged across six math test sets and reported at various training steps.

# A APPENDIX

## A.1 ABLATION STUDY

**Analysis of reasoning behavior.** We analyze the reasoning behavior of the models before and after applying Rollout Response Recomposition (RoRecomp), as shown in Tab. 10. Inspired by ThinkPrune Hou et al. (2025), we divide the reasoning process into three phases: problem-understanding, problem-solving, and self-verification. Each phase may consist of multiple reasoning steps, with each step separated by double newlines ("\n\n"). We use DeepSeek-V3-0324 Liu et al. (2024) to assign each reasoning step to its corresponding phase, then count the number of steps and tokens for each phase.

The results (Tab. 10) demonstrate that RoRecomp leads to a substantial reduction in both the number of steps and tokens across all reasoning phases, with the most pronounced effect observed in the self-verification phase. For example, on the DeepSeek-R1-7B model, RoRecomp reduces the number of self-verification steps by 88.6% and the corresponding token count by 83.6%. Similar trends are observed for the DeepSeek-R1-1.5B model. This suggests that lengthy self-verification is largely redundant and can be significantly streamlined without compromising performance.

In contrast, the reduction in the problem-understanding phase is more modest, with the number of steps decreasing by less than 42% for both model sizes. Notably, RoRecomp also changes the distribution of steps and tokens among the three reasoning phases. In the original models, self-verification consumed more tokens than problem-understanding (e.g., 1,320 vs. 474 tokens for DeepSeek-R1-7B), whereas after applying RoRecomp, problem-understanding takes up more tokens. This shift indicates that RoRecomp encourages the model to focus more on understanding the problem, while reducing unnecessary elaboration during self-verification.

**Ablation study on the compensation batch $\mathcal{B}_{\mathbf{comp}}$.** We investigate the effects and training strategy of $\mathcal{B}_{\text{comp}}$ separately. Specifically, $\mathcal{B}_{\text{comp}}$ is used to preserve the model's exploration capacity and prevent the policy model from overfitting to a narrow subset of the response distribution. In our experiments (Fig. 4), we compare a setting where compensation batches are discarded and only priority batches are used for training (denoted as "wo/ $\mathcal{B}_{\text{comp}}$") with RoRecomp, which leverages both priority and compensation batches. As shown in the response length curves on the test set, "wo/ $\mathcal{B}_{\text{comp}}$" exhibits a rapid decrease in response length during the initial training phase, whereas RoRecomp achieves a smoother reduction. In terms of accuracy, "wo/ $\mathcal{B}_{\text{comp}}$" suffers a sharp drop after 80 steps, ultimately reaching 42%, which is 6% lower than RoRecomp. These experimental results demonstrate that the

compensation batch is indispensable; otherwise, the model's exploration space would be damaged, leading to a significant drop in performance.

## A.2 THE USE OF LARGE LANGUAGE MODELS

We utilize an LLM to assist with paper editing and correcting grammatical errors.

