# OpenReview forum: "RoRecomp: Enhancing Reasoning Efficiency via Rollout Response Recomposition in Reinforcement Learning"
_ICLR.cc/2026/Conference — Submitted to ICLR 2026_

### Official Review · Reviewer_LyEV · 2025-10-27

**Soundness:** 2
**Presentation:** 2
**Contribution:** 2
**Rating:** 2
**Confidence:** 3

**Summary:**

This paper proposes a method called RoRecomp to improve the reasoning efficiency of LLMs under the RLVR framework, i.e., reducing verbosity in the reasoning process while maintaining accuracy.

**Strengths:**

- Simple, practical, and plug-and-play: it does not change the reward or the main RL algorithm; it only “recomposes batches” of training samples after rollout.
- Experiments show that it significantly reduces length while achieving performance comparable to GRPO.

**Weaknesses:**

- The experimental setup lacks details. For example, it is not clearly stated how many samples per question RoRecomp uses, what value α is set to, etc. Baseline details are also missing, such as how many samples per question GRPO uses.
-  The pipeline  of this method isn't very clear. Will the total batch size be reduced at each step? Are the responses in the replay buffer only used after sampling the same question multiple times ? Will this outdated data affect model training?

**Questions:**

- The paper introduces $p_\text{comp}$, but its exact purpose is not clearly explained. It mentions “Once the buffer is full, the oldest experiences are popped to form a compensation batch B_comp for an additional training step.” Then what is the role of $p_\text{comp}$?
- How is the case handled when a question’s responses are all correct or all incorrect?
- Does RoRecomp require sampling more responses per question? Is there a comparison of training time with methods like GRPO?

---

### Official Review · Reviewer_ykUR · 2025-10-28

**Soundness:** 3
**Presentation:** 3
**Contribution:** 2
**Rating:** 4
**Confidence:** 3

**Summary:**

This paper introduces RoRecomp, a reinforcement learning framework designed to enhance reasoning efficiency in large language models. Instead of modifying reward functions, RoRecomp strategically reorganizes rollout data into priority batches—short-correct and long-incorrect responses—to directly bias optimization toward concise, accurate reasoning. Complementary compensation batches stabilize training via replay. Experiments across zero RL training, agentic RL, and thinking compression show substantial reductions in reasoning length (up to 74%) and tool calls (46.8%) with minimal or no performance degradation.

**Strengths:**

The proposed method itself is coherent and easy to follow. RoRecomp is easy to integrate with PPO or GRPO frameworks without altering the reward function. The authors conduct experiments on several benchmarks to validate the effectiveness of the proposed method.

**Weaknesses:**

While RoRecomp presents an effective way to improve reasoning efficiency, several limitations temper its overall contribution.

**(1) Limited conceptual novelty.** The method builds upon existing ideas such as prioritized experience replay and curriculum-based sample selection, repurposing them for RL with verifiable rewards. Although the empirical improvements are impressive, the innovation primarily lies in application design rather than in proposing a fundamentally new optimization principle.

**(2) Lack of theoretical justification.** The paper provides little theoretical analysis explaining why recomposing batches in this specific manner leads to improved efficiency. There is no formal connection established between the altered sampling distribution and reduced gradient variance or faster convergence. This weakens the claim that the method systematically improves optimization stability.

**(3) Reliance on heuristics.** Key design choices, such as the α selection ratio and compensation batch scheduling, are tuned empirically without clear guidelines. The balance between brevity and stability may not generalize across model scales or domains.

**(4) Limited scalability and domain coverage.** Experiments are conducted mostly on reasoning-centric tasks and models up to 8B parameters. It remains uncertain whether RoRecomp maintains its benefits for larger models, open-domain tasks, or multi-modal reasoning scenarios where efficiency manifests differently.

**Questions:**

1.	How sensitive are results to the α threshold and replay scheduling scheme?
2.	Have the authors evaluated scalability on models larger than 30B in RL settings?

---

### Official Review · Reviewer_Z6u4 · 2025-10-28

**Soundness:** 3
**Presentation:** 3
**Contribution:** 2
**Rating:** 4
**Confidence:** 3

**Summary:**

This paper introduces RoRecomp to improve the efficiency of reasoning in LLMs trained with RLVR. The core idea is to recompose training batches after the rollout stage, separating responses into priority batches and compensation batches.  Empirical results across three settings—zero RL training, agentic RL, and hinking compression—demonstrate that RoRecomp significantly reduces reasoning length, lowers redundant tool use.

**Strengths:**

RoRecomp introduces a simple yet powerful data recomposition strategy that enhances reasoning efficiency in reinforcement learning without altering rewards. The method is broadly applicable across settings, theoretically grounded, and empirically validated with extensive experiments showing large reductions in reasoning length and tool use while maintaining performance.

**Weaknesses:**

**1.Limited novelty.**
The idea of recomposing rollout batches resembles reward shaping and sampling bias control, making it unclear whether the conceptual contribution is substantially new beyond existing variance-reduction or data-selection techniques.

**2.Blurry conceptual boundary.** The method is described as “orthogonal to reward shaping,” yet experiments include truncation penalties and length limits, which effectively act as implicit reward shaping, weakening the claimed orthogonality.

**3.Insufficient comparison with strong baselines.** Important RL reasoning baselines such as DAPO, RLOO, or VAPO are missing, which limits the strength of empirical validation.

**4.Scalability and group-size effects underexplored.** Since the motivation centers on small rollout groups causing high variance, it would be informative to study how performance scales when group size changes or when applied to larger distributed setups

**Questions:**

see Weaknesses

---

### Meta-Review · Area_Chair_1xfA · 2026-01-08

**Summary:**

All negative scores and no rebuttal.

**Reviewer Concerns:**

All negative scores and no rebuttal.

**Reviewer Scores:**

All negative scores and no rebuttal.

---

### Decision · Program_Chairs · 2026-01-26

Reject